# What Is the State of Development of Eco-Wellbeing Performance in China? An Analysis from a Three-Stage Network Perspective

Yu Zhang [1], Xi Cai [1,*], Yanying Mao [2], Liudan Jiao [1] and Liu Wu [1]

1   School of Economics and Management, Chongqing Jiaotong University, Chongqing 400074, China; yuzhang@cqjtu.edu.cn (Y.Z.); jld@cqjtu.edu.cn (L.J.); 990202200030@cqjtu.edu.cn (L.W.)
2   Department of Communication Engineering, Chongqing College of Electronic Engineering, Chongqing 401331, China; d220201024@stu.cqupt.edu.cn
*   Correspondence: 622210121005@mails.cqjtu.edu.cn

**Abstract:** After rapid development in recent decades, China has laid a strong economic foundation and established material conditions. Despite this, the country still confronts a range of challenges that impede higher-quality economic and social development. To measure high-quality regional development, eco-wellbeing performance (EWP) has become an important tool that seeks to strike a balance between economic development, human wellbeing, and environmental protection. This paper proposes a three-stage network efficiency evaluation model to decompose EWP into three stages, namely ecological economic efficiency (EEE), economic innovation efficiency (EIE), and innovation wellbeing efficiency (IWE). A directional distance function (DDF) model was utilized to assess the overall EWP efficiency and phase-in efficiency for 248 cities over the period from 2016 to 2020. The study results indicate that EWP in China is generally low. In terms of the three sub-stages of EWP, the first stage performs optimally, the third stage performs the second best, and the second stage is the worst. The research suggests that the principal reason for the generally low EWP may be linked to the efficiency of the transformation of economic investments into innovative outcomes in the second stage. From a regional viewpoint, EWP generally shows a gradually decreasing trend from the east, central, and west to the northeast, but the stage performance varies among regions. The eastern region has lower EEE, but both EIE and IWE perform better than the national average; the central region is well-balanced between the three stages; the western region leads the country in EEE, but lags in both EIE and IWE; and the northeast region has inferior performance in all stages. This study's findings provide vital reference values for policy-makers to determine key points for enhancing EWP in different regions of China.

**Keywords:** eco-wellbeing performance; three-stage network model; directional distance function; regional differences





## 1. Introduction

Today's world is undergoing significant and unprecedented changes. Global climate change, social unrest due to the growing inequality between certain countries, and the impact of the fourth technological and industrial revolution on the labor market are all severe obstacles to humanity's goal of a better life [1]. At the same time, China, as the world's most populous country, has quietly changed its main social contradictions, and its unbalanced and inadequate development can no longer meet the people's growing demand for a high quality of life [2]. Providing a better ecological environment, high-quality public services, safeguarding people's welfare, and achieving sustainable development will be important elements in satisfying people's demands for a wonderful life. After over 40 years of rapid growth through reform and opening up, China has laid a good financial foundation and material conditions. However, its development still faces a series of challenges, some of which constrain China from achieving higher quality and a more sustainable society.

Environmental degradation poses a major challenge in China today. Continuous economic development, since its implementation of reform and openness, has exerted considerable stress on the ecological surroundings, energy, and resources. As observed, the Soil Pollution Prevention and Control Action Plan, which was reissued in 2016, expresses China's worrisome situation regarding soil pollution [3]. Furthermore, 70.7% of China's 338 urban centers, at the prefecture level and higher, had inadequate ambient air quality in 2017 [4]. The total volume of China's water resources was only 2876.12 billion cubic meters in 2017, indicating a per capita possession of 2075 cubic meters, which represents just one-third of the world's per person possession in 2014 [5]. Public service remains to be another major challenge for China. Initially, in 1978, China's urban population was under 20%, however, by late 2018, the urbanization rate for the resident population was 59.6%, showing a large increase of 40 percentage points [6]. As more residents migrate to cities, resulting in soaring demand for public services, this growing demand exceeds the limits of urban public resource services. China's economic growth is also a crucial challenge for the future. From 1978 to 2017, GDP experienced an average annual growth of 9.5% at fixed prices which is significantly faster than the world economy's average annual increase of about 2.9% over the same time, positioning it among the world's major economies. However, since 2015, China's economic growth has slowed, with a yearly average of less than 7% [7].

The Chinese government has introduced a variety of policies and initiatives to address the challenges, including the promotion of social equality and environmental protection. In 2017, the State Council released the "13th Five-Year Plan for Promoting Equalization of Basic Public Services". The plan aims to achieve a more equitable and better basic public service system by 2020 [8]. The following year, the Chinese government introduced the Three-Year Action Plan for Winning the BlueSky Battle [9]. The plan intends to mitigate air pollutants and greenhouse gas emissions by reorganizing the industrial layout, energy, transportation, and land use structure. Since the 13th Five-Year Plan, China has voluntarily forsaken its rapid economic growth strategy to promote a more sustainable economic outlook [10].

Despite various central government policies on ecology and public services, there is still an open question of whether municipal governments have taken effective action, and more significantly, whether these measures help to improve the wellbeing of people. Eco-wellbeing performance (EWP) seeks to balance economic progress with environmental protection and social welfare by prioritizing the sustainable exploitation of natural resources and the protection of the earth and is an effective indicator of the ability of regions to develop sustainably [11,12]. As a result, EWP has gained increasing popularity in China [13–17], and the state of EWPs in China has profoundly influenced the process of sustainable development in the world.

However, there are some shortcomings in the existing studies. Firstly, most studies are stuck in a one- or two-stage research paradigm. Due to this limitation, natural resource consumption, economic growth, science and technology innovation, and people's wellbeing have not been well linked to measuring the stages of sustainable development in urban areas. Secondly, most studies on EWP have focused on the national or provincial level, and due to the overly large data compilation and collection, very little research has been conducted on prefecture-level cities in China.

Therefore, to integrate natural resource consumption, economic growth, science and technology innovation, and people's welfare into the same urban system, this paper innovatively decomposes EWP into three interrelated stages, including ecological economic efficiency (EEE), economic innovation efficiency (EIE), and innovation wellbeing efficiency (IWE). On this basis, a three-stage network model is proposed in the text, incorporating the above processes into the same framework. The directional distance function (DDF) will be utilized to measure the overall efficiency in 248 cities in China from 2016 to 2020, as well as the efficiency of each phase. By identifying the efficiency of urban areas at different stages

of EWP, this paper aims to identify specific areas for policy enhancement and formulate strategies to address existing challenges.

The rest of the paper is organized as follows. Section 2 provides a literature review. Section 3 describes the methodology, study area, indicators, and data sources of this paper. Section 4 provides an analysis of the empirical results. Section 5 presents a brief discussion of the results. Section 6 gives the conclusion of this study.

## 2. Literature Review

Are resource consumption, economic growth, technological innovation, and people's wellbeing linked in urban systems? How are they linked? Is it possible to combine them to measure the sustainability of cities? These are the questions that are of interest in this paper. Three existing concepts, eco-efficiency, innovation efficiency, and eco-wellbeing performance, are indispensable in clarifying the transformational relationship between these four aspects.

### 2.1. Eco-Efficiency

The notion of eco-efficiency was proposed in 1992 by the World Business Council for Sustainable Development (WBCSD), to create more worth using fewer resources while minimizing the environmental effect of economic activities by optimizing natural resource use, minimizing waste and pollution [18]. Following the adoption of China's sustainable development policy, the focus of eco-efficiency studies shifted rapidly from provinces to city clusters, and ultimately to prefecture-level cities. In provincial eco-efficiency studies, key areas of concern entail industrial restructuring, technological innovation, improving resource utilization efficiency, and promoting sustainable development. These studies aimed to measure eco-efficiency levels and investigate relationships between variables such as industrial structure, technological innovation, and resource use efficiency in individual provinces [19–21]. For example, Zhou et al. [22] estimated the eco-efficiency levels in Guangdong Province and found that the average eco-efficiency was low, mainly attributable to low resource use efficiency and unfavorable industrial structure. However, technological innovation had a beneficial impact on eco-efficiency levels in Guangdong Province. In urban clusters, the research focus is on urbanization and regional coordination. Several studies aim to determine the best allocation of resources between different regions to achieve regional eco-efficiency [23–25]. For instance, Ren et al. [26] evaluated the efficiency in urbanization and eco-efficiency levels in seven urban agglomerations in the Yellow River Basin of China between 2006 and 2019 and found a significant spatial correlation between the two variables. Thus, effective policies and interventions are needed to achieve the coordinated development of eco-efficiency and urbanization. In cities, the research focus revolves around green urbanization, low-carbon development, and sustainable urban development. Studies have investigated various eco-efficiency indicators, such as carbon emissions, energy consumption, and water use, concerning assessing eco-efficiency levels of different prefecture-level cities in China [27–30]. For instance, Liu et al. [31], Bai et al. [32], and Ren et al. [33] all examined eco-efficiency levels in prefecture-level cities in China by evaluating resource consumption, economic output, and pollution emissions.

### 2.2. Innovation Efficiency

Innovation efficiency is another crucial indicator of the sustainable development potential of a nation or area [34]. According to Vollebergh and Kemfert [35], sustainable development is not feasible without technological innovation and progress. Innovation plays a crucial role in the development of China's green economy while driving the country's economic, ecological, and social sustainability, as emphasized by Liu and Dong [36] and Ke et al. [37]. As such, innovation efficiency in China has been a popular research subject, with scholars investigating it from various viewpoints. Studies have revealed that innovation efficiency in many Chinese cities is generally low and polarized, with a noticeable spatial aggregation effect [38,39]. In contrast, a few scholars have found the in-

novation efficiency of Chinese urban areas to be on the rise, even though some urban areas in the northeast, northwest, and southwest regions are an exception to this trend [40,41]. Nonetheless, the regional disparity in green innovation efficiency among Chinese cities is undeniably narrowing, as indicated by Zhao et al. [42] and Liao and Li [43], which suggests the gradual emergence of a spatially synergistic innovation division of labor with "R&D in the East + transformation in the in the Southwest and Northeast". Effective environmental regulation is an essential factor that could enhance innovation efficiency in China, according to Zhang et al. [44] and Fan et al. [45]. Meanwhile, external factors, such as R&D financing, knowledge diversity, and transport infrastructure, also boost innovation efficiency, leading to the sustainable development of the region [46–48]. Additionally, urban innovation is not solely dependent on local innovation activities but also on the geographical position of cities in their intercity co-invention network [49,50].

### 2.3. Eco-Wellbeing Performance

Eco-wellbeing performance (EWP) was introduced by Daly [51] in 1974 and defined by Zhu et al. [52] as the enhancement of human welfare per unit of ecological depletion. EWP upgrades eco-efficiency, bringing it closer to the goal of sustainable development [53,54]. Many scholars have studied EWP in different cities and regions of China. Some studies have focused on the space-time evolution and drivers of EWP in Chinese cities [55,56]. Scholars generally agree that EWP is low in Chinese cities and performs better in the east than in the center and west [11,57]. However, the EWP of cities is now improving [11]. Specific regions, for instance, the Yangtze River Delta [58], the Middle Yangtze River Urban Agglomeration [59], the Yellow River Delta [15], and Beijing-Tianjin-Hebei Agglomeration [60], have also become popular targets for EWP studies. At the inter-provincial level, studies indicate that the general level of EWP in China is declining, and the spatial gap may increase further [16,17,61]. In contrast, Deng et al. [13] reported that EWP improved nationally, with greater improvements in the east.

### 2.4. The Connection of Ecology, Economy, Innovation, and Wellbeing

All of the above studies take ecological inputs to cities as a starting point and measure their efficiency in translating into economic output, innovation output, and people's wellbeing, respectively. However, the correlation between the economy, innovation, and people's wellbeing has been ignored. What is certain is that innovation efficiency is an upgrade of eco-efficiency and eco-wellbeing performance is an upgrade of innovation efficiency. The emergence of innovation efficiency has led to a shift away from a focus on crude economic growth, and the introduction of eco-wellbeing has anchored the purpose of development in human beings themselves. Today, there is a consensus that cities are complex mega systems [62]. Therefore, the depletion of natural resources, economic development, technological innovation, and the wellbeing of people in cities can never exist separately from this system. It was not until Kiani Mavi et al. [63] used the two-stage network DEA to measure eco-efficiency and eco-innovation that a new way of thinking was opened to clarify this connection. Since then, many scholars have started to conduct studies using this approach, for example, Hou et al. [64] and Xia and Li [60] decomposed EWP into ecological economic efficiency and economic welfare efficiency. However, they usually only remain in a two-stage research paradigm [65–68]. The closest to the idea of this study is that of Zhang et al. [69], who measured eco-efficiency, eco-technological innovation, and eco-welfare performance in 102 countries, however, they still failed to link all three into the same framework. The final objective of urban development is the welfare of the people, while technological progress and innovation are only an intermediate process. Therefore, this paper innovatively decomposes EWP into three interrelated stages, namely, the conversion of ecological inputs into economic outputs, economic inputs into innovation outputs, and innovation inputs into welfare outputs. For ease of presentation, they are denoted by ecological economic efficiency (EEE), economic innovation efficiency (EIE), and innovation wellbeing efficiency (IWE), respectively.

The purpose of this paper is to construct an appropriate set of models and indicators to measure the overall and stage efficiency of EWP in 248 Chinese cities for the period 2016–2020. By analyzing the performance of each sub-stage, the key aspects affecting the EWP are identified, and thus specific areas for policy improvement are suggested.

## 3. Methodology

### 3.1. Network-DDF

Traditional data envelopment analysis (DEA) usually treats the decision unit as a whole, focusing only on the initial inputs and final outputs, and the intermediate processes within the system are often ignored [58,64]. Färe and Grosskopf [70] proposed network DEA to address this problem by using the output variables of the previous stage as input variables for the next step and opening a "black box" by connecting adjacent locations in the system with one or more variables. In this way, the efficiency evaluation problem of multistage decision systems can be effectively solved. According to this idea, the three-stage network of EWP constructed in this paper is shown in Figure 1.

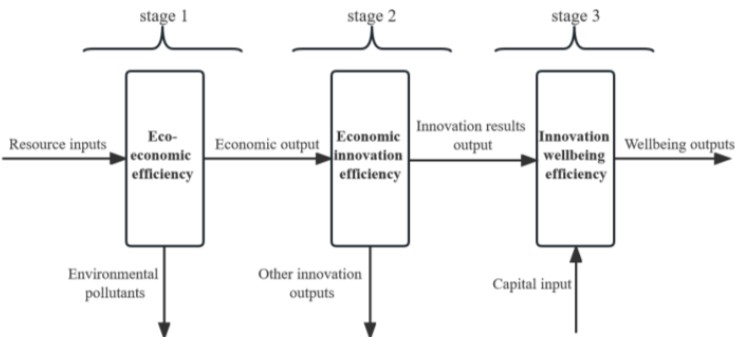

**Figure 1.** Three-stage network framework.

Although there are many methods to measure efficiencies, such as the ratio method and stochastic frontier analysis, DEA has become the mainstream method nowadays due to its advantages of handling multiple inputs and outputs and not requiring pre-set parameters [54,60]. Meanwhile, among various DEA extension models, radial models, slack-based measure (SBM), and directional distance function (DDF) are the three most widely used ones. However, the radial DEA model cannot directly deal with undesired outputs and therefore cannot effectively balance the two objectives of minimizing undesired outputs and maximizing desired outputs [11]. Although the SBM model can deal with undesired outputs well, it has the characteristic of better and more input and output indicators [71], which means that it tends to underestimate the efficiency in the actual situation of this study. Chung et al. [72] proposed the DDF model which allows the simultaneous reduction in inputs and non-desired outputs as well as increasing desired outputs without underestimating efficiency in the case of less desired outputs. Therefore, in this paper, the DDF is used to measure the overall and phased efficiency of the three-stage network. The model is constructed as follows:

Suppose a production system has n decision units, each of which has m inputs and produces d desired outputs and u undesired outputs. Define the vectors $X = (x_1, x_2, \ldots, x_n) \in R_+^{m \times n}$, $Y = (y_1, y_2 \ldots, y_n) \in R_+^{d \times n}$, $b = (b_1, b_2 \ldots, b_n) \in R_+^{u \times n}$ as input, desired output, and undesired output variables, respectively. Assuming that the DMU is the decision unit to be estimated, the production possibility set is: $P^t(x) = \{(y, b) : x \text{ can produce } (y, b)\}$. It satisfies two assumptions: one is that the desired output $y$ is correlated with the undesired output b. The second is that non-desired outputs are weakly disposable, meaning that a decrease in some outputs leads to a decrease in other outputs, or that an increase in some

outputs must ensure that other outputs increase at the same time in some proportion. On this basis, we introduce the directional distance function:

$$\overrightarrow{D}_0(x,y,b;g_y - g_b) = sup\{\theta : (y + \theta g_y, b - g_b) \in P(x,y,b)\} \tag{1}$$

The directional distance can be solved by the following planning:

$$\begin{aligned}
&\overrightarrow{D}_0^t\left(x^{tk}, y^{tk}, b^{tk}, g_y^{tk} - g_b^{tk}\right) = max\theta \\
&s.t. \sum_{j=1}^{n} \lambda_j y_{rj}^t \geq (1+\theta) y_{rk}^t, r = 1, 2, \ldots, d \\
&\quad\ \sum_{j=1}^{n} \lambda_j b_{lj}^t = (1-\theta) b_{lk}^t, l = 1, 2, \ldots, u \\
&\quad\ \sum_{j=1}^{n} \lambda_j x_{ij}^t \leq (1-\theta) x_{ik}^t, i = 1, 2, \ldots, m \\
&\quad\quad \lambda_j \geq 0, j = 1, 2, \ldots, n
\end{aligned} \tag{2}$$

where $(g_y - g_b)$ is the direction vector of desired output increase and undesired output decrease. $\theta$ denotes the distance between the actual output value at point $(y, b)$ and its projection point $(y + \theta g_y, b - \theta g_b)$ on the production frontier plane, i.e., the maximum possible proportion of $y$ increase and $b$ decrease at the same time, the greater the value, the smaller the efficiency, indicating that the evaluation unit has more room for desired output increase and pollution decrease; conversely, the greater the value, the higher the efficiency. When $\theta = 0$, it implies that the evaluation unit is at the production frontier surface and its production is valid.

### 3.2. Indicator System

It is essential to build a suitable system of assessment indicators to accurately evaluate the efficiency of EWP and its sub-phases using the above-mentioned methods. In this paper, a complete set of evaluation indicators is constructed according to the principles of scientific and availability of indicator data, as shown in Table 1. To eliminate the effect of inflation, GDP, science and technology expenditure, public budget, and wage income are converted to constant prices using 2016 as the base year. To eliminate the effects of differences in size and population across cities, per capita or average values were used for each indicator data.

**Table 1.** Evaluation index system of EWP.

| Phase | Category | Indicators | Sources |
|-------|----------|------------|---------|
| Stage 1 | Input | Per capita water | [29] |
| | | Per capita electricity | [60] |
| | | Per capita built-up area | [28] |
| | Desired output | Per capita GDP | [56] |
| | Undesired output | Per capita wastewater | [22] |
| | | Per capita $SO_2$ | [28] |
| | | Per capita industrial fume and dust | [31] |
| Stage 2 | Input | Per capita GDP (**intermediate variable**) | |
| | Output | Science and technology expenditure per capita | [68] |
| | | Number of patents granted per million people | [44] |
| | | Number of general higher education teachers per million people | [73] |

**Table 1.** *Cont.*

| Phase | Category | Indicators | Sources |
|---|---|---|---|
| Stage 3 | Input | Number of patents granted per million people (**intermediate variable**)<br>Public budget per capita (**additional input**) | [11] |
| | Output | Number of college students per $10^5$ people<br>Number of doctors per $10^5$ people<br>Average wage of employees | [11]<br>[11]<br>[59] |

### 3.3. Study Area

The study area of this paper covers 248 prefecture-level cities in mainland China. China is a large country with a massive population and a vast land area, and different regions have large differences in various aspects such as natural ecology and social environment. For subsequent analysis, this paper adopts a spatial perspective and divides mainland China into four major zones according to the distribution of natural resources and economic and social development: the eastern, central, western, and northeastern areas [1]. As shown in Figure 2: 10 regions, such as Beijing and Tianjin, belong to the east; 6 regions, such as Hubei and Hunan, belong to the center; 12 regions, such as Sichuan and Chongqing, belong to the west; and the northeast contains the 3 provinces of Liaoning, Jilin, and Heilongjiang.

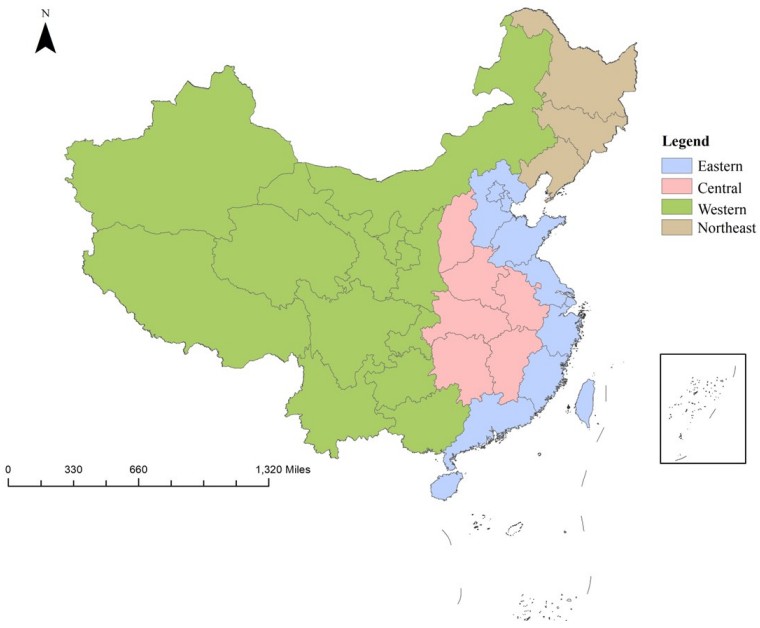

**Figure 2.** Four major regions in China.

### 3.4. Data Collection

The data corresponding to the above indicators are from prefecture-level cities in China. As of 2020, there are 338 prefecture-level cities in China, but due to missing or unavailable data for some cities, we only selected 248 cities for empirical analysis. Since the latest data from the Statistical Yearbook are only updated to 2020, we chose the 5 years of 2016–2020 as the study period. These five years are exactly the thirteenth five-year plan for China's national economic and social development, which has important research value. All data for 248 prefecture-level cities from 2016–2020 were obtained from the China Urban Statistical Yearbook, the China Urban Construction Statistical Yearbook, the State Intellectual Property Office, and local statistical yearbooks of provinces and cities.

## 4. Analysis of the Results

Using the model and data presented in Section 3, with the help of the computer program MAXDEA, it was possible to obtain the overall efficiency and sub-stage efficiency values of the EWP for each city from 2016 to 2020.

### 4.1. Overall Analysis of the EWP

The obtained overall efficiency of EWP for 248 cities for 2016–2020 is plotted as a scatter plot, as shown in Figure 3, where the scatter points represent the different cities and the black horizontal line indicates the mean value of the overall efficiency of EWP for all cities in that year, and the number of effective decision units with an efficiency value of 1 is circled. From the figure, it can be observed that the mean value of EWP for each year from 2016 to 2020 remains at a lower level of around 0.25. Moreover, a large number of cities are clustered in the low-level area below the mean value line. Only a small number of cities reach the DMU effectively: only one city in Bozhou in 2016, only one city in Shenzhen in 2018, two cities in Shenzhen and Zhumadian in 2019, and no cities reached the DMU in 2017 and 2020.

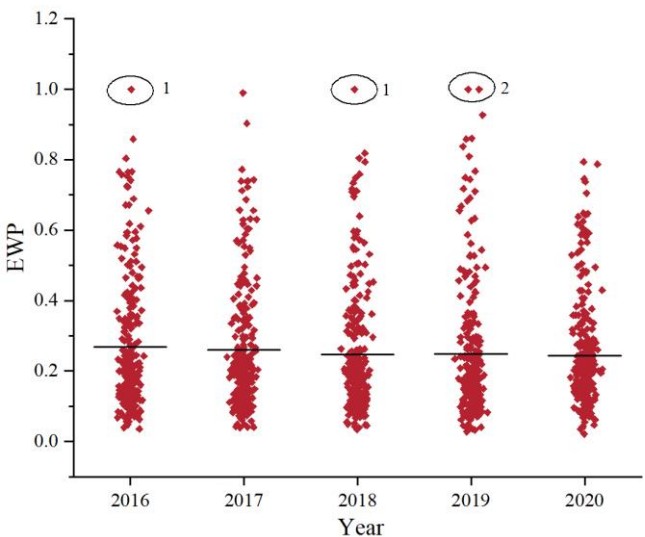

**Figure 3.** Scatter plot of overall EWP efficiency for 248 cities from 2016–2020.

The low level of overall EWP efficiency confirms that China's rapid economic development has also put an enormous burden on the ecological environment, energy, and resources. The ability of cities such as Shenzhen and Bozhou to maintain a balance between economic growth and human wellbeing amidst the tide of reform is well worth learning from other cities. Shenzhen, as the first special economic region in China, has been at the leading edge of reform and opening up and has taken the lead in forming a modern industrial system with a ladder of emerging industries, future industries, modern service industries, and advantageous traditional industries. Bozhou City, located in Anhui Province, has long been known as the capital of health care. In recent years, they have devoted themselves to the development of special advantageous industries, for instance, traditional Chinese medicine and liquor according to local conditions, which have created a powerful impetus for the development of Bozhou.

Using ArcMap software, the average values of the overall EWP efficiency of 248 cities were plotted on a map of China, as shown in Figure 4a. For easy observation, all cities were divided into three groups, which were represented by different colors: Group 1 (EWP ≤ average), Group 2 (average < EWP ≤ 0.5), and Group 3 (EWP > 0.5). In addition, the average values of the overall EWP efficiency in the four major areas of China for 2016–2020 are plotted as radar plots (Figure 4b). As seen in Figure 4a, cities in Group 1 and Group 3 are more dispersed, while Group 2 cities are more clustered in the eastern

and central regions. Specifically, the higher-level EWPs are mostly clustered in the Yangtze River Delta, Pearl River Delta, Yellow River Delta, and Beijing-Tianjin-Hebei city clusters. Thus, EWPs form a gradually decreasing pattern in the eastern, central, western, and northeastern areas, which is more clearly reflected in Figure 4b.

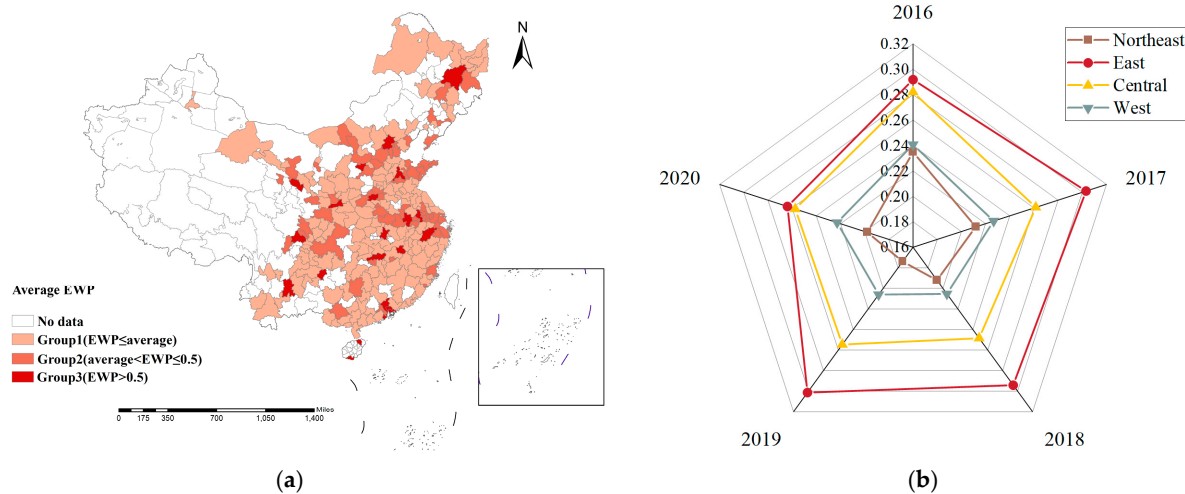

**Figure 4.** Spatial distribution characteristics of the overall EWP efficiency: (**a**) Distribution of the EWP mean on the map of China; (**b**) Evolution of the EWP mean in the four main regions.

Due to natural and historical reasons, the northeast and eastern regions are more economically and socially developed, the central region is in the middle, and the western region is less developed. However, in the regional comparison of EWP in recent years, the northeast is always at the bottom of the four regions. As an old industrial base, Northeast China has been relying on resource development, heavy industry, and a highly energy-consuming industry to sustain its economic development for a long period. The massive consumption of natural resources and the destruction of the ecosystem has led to it being at the bottom of the EWP today. The Yangtze River Delta, Pearl River Delta, Yellow River Delta, and Beijing-Tianjin-Hebei have been the leaders of China's economic and social development due to their strong financial strength and advantageous geographical location.

### 4.2. Stage Analysis of EWP

The causes of the general inefficiency of EWPs are the focus of this study. Therefore, a three-stage network is used to open up three sub-stages within the EWP system, which are eco-economic efficiency, economic innovation efficiency, and innovation wellbeing efficiency. The performance of the three sub-stages is analyzed to find the key points to enhance EWP.

#### 4.2.1. Stage 1: Ecological Economic Efficiency

The first stage of the internal transformation of EWP is the conversion of ecological inputs into economic outputs, i.e., ecological economic efficiency (EEE). Similar to previously, the EEE of 248 cities for 2016–2020 was plotted as a scatter plot, as shown in Figure 5. As can be seen from the figure, the mean value of EEE was maintained at a high level between 0.6 and 0.8, with a slight decrease in 2018–2020. Up to 103 cities reached the effective level in 2016 and 2017, and 67, 58, and 50 cities reached the effective level in 2018–2020, respectively.

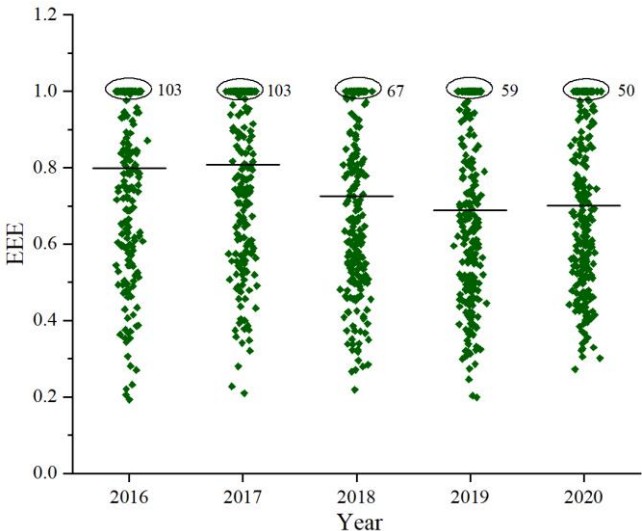

**Figure 5.** Scatter plot of EEE for 248 cities from 2016 to 2020.

As we all know, since the reform and opening up, the environmental challenges faced by China have become more and more severe against the background of rapid economic expansion and sustained population increase. The environmental pressure has pushed the Chinese government to rethink the traditional development model, and the notion of ecological civilization has emerged. In 2012, China integrated the construction of ecological civilization into the overall layout of "Five-in-One" and aspired to become an essential contributor to the construction of global ecological civilization and contribute "Chinese wisdom" to solve global environmental problems. Since then, the Chinese government has positively pushed for the ecological transformation of traditional industries, the industrialization of ecological protection, and the greening of the economic and social development system. In addition, the supply-side structural reform and the energy production and consumption revolution have fundamentally changed China's national economic and social development system. Meanwhile, the structure within the industry is also being optimized. High-tech industries, high-end manufacturing industries, internet finance, and other new industries are developing rapidly, and the level of resource conservation and environmental friendliness is rising. The green transformation of production models and industrial structures also aims to change everyone's lifestyle, making low-carbon travel, green communities, and eco-cities the keynote of social development. The 2016–2020 high-level EEE shows that China's ecological environment has been improved and is moving towards a green development path.

Similar to previously, using ArcMap software, the EEE averages of 248 cities were plotted on a map of China, as in Figure 6a. All cities are classified into three groups, which are indicated by different colors: Group 1 (EEE ≤ average), Group 2 (average < EEE ≤ 0.9), and Group 3 (EEE > 0.9). In addition, the average values of EEE for 2016–2020 in the four major regions of China are plotted as radar plots (Figure 6b). From Figure 6a, it can be observed that EEE is significantly better in Western and Central China than in Eastern and Northeastern China. This is more clearly reflected in Figure 6b, where EEE is in descending order: western, central, eastern, and northeastern.

In 2000, China began to execute the Western Development Strategy to promote the accelerated growth of the western region. Since then, the economic level of the western region has been rising rapidly, with the average annual increase in income of urban and rural residents exceeding 10%, higher than the national and eastern averages. In addition, China has carried out a lot of pioneering work on ecological and environmental management in the western region and has achieved fruitful results. The vegetation cover of forests and grasslands has increased significantly, the Great Desert has been effectively managed, and the severe flood control situation in the lower reaches of the Yellow River has been

fundamentally changed. The ecological environment of the central region has also been greatly improved due to the comprehensive integrated development strategy of the central and western areas.

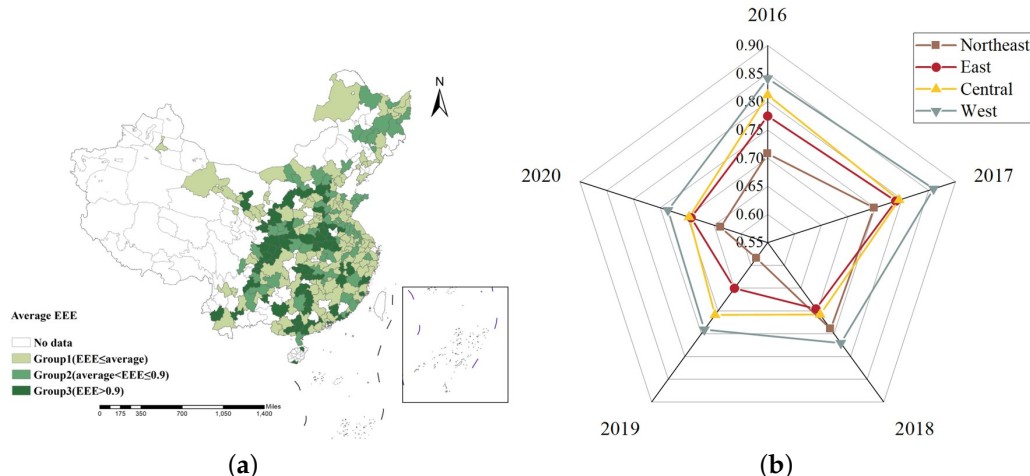

**Figure 6.** Spatial distribution characteristics of EEE: (**a**) Distribution of the EEE mean on the map of China; (**b**) Evolution of the EEE mean in the four main regions.

### 4.2.2. Stage 2: Economic Innovation Efficiency

The second stage of the internal transformation of EWP is the conversion of economic inputs into innovation outcomes, i.e., economic innovation efficiency (EIE). The EIE of 248 cities for 2016–2020 is plotted as a scatter plot shown in Figure 7. From Figure 7, it can be seen that the EIEs in 2016–2020 were all at a low level of about 0.2. The cities that reached the effective level in 2016–2019 are divided into 5, 1, 4, and 5, and no cities reached the effective level in 2020.

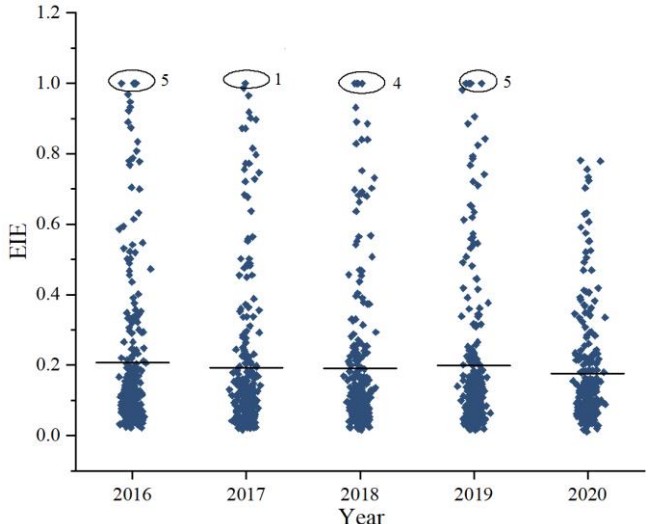

**Figure 7.** Scatter plot of EIE for 248 cities from 2016 to 2020.

In the new wave of scientific and technological revolution and industrial change, knowledge innovation, technological innovation, and industrial innovation are deeply integrated, and basic research is increasingly becoming an important source and driving force for the scientific and technological revolution and industrial change. The proportion of basic research investment to total R&D investment in China is still at a relatively low position. In 2016, for example, China's fundamental research investment accounted for

only 5.2% of R&D expenditures, far below the level of 15–25% of the world's major innovative countries. From the innovation index released by the World Intellectual Property Organization, China's science and technology innovation still lags far behind high-income economies such as the United States, Japan, and Germany, with the major shortcomings being high-quality patents as well as high-quality papers.

The average values of EIE for 248 cities were plotted on a map of China, as shown in Figure 8a. All cities are classified into three groups, which are indicated by different colors: Group 1 (EIE ≤ average), Group 2 (average < EIE ≤ 0.5), and Group 3 (EIE > 0.5). In addition, the average values of EIE for 2016–2020 in the four major areas of China were plotted as radar plots (Figure 8b). Observing Figure 8, it can be seen that the highest EIE level was in the eastern region, followed by the central region, and finally the western and northeastern regions.

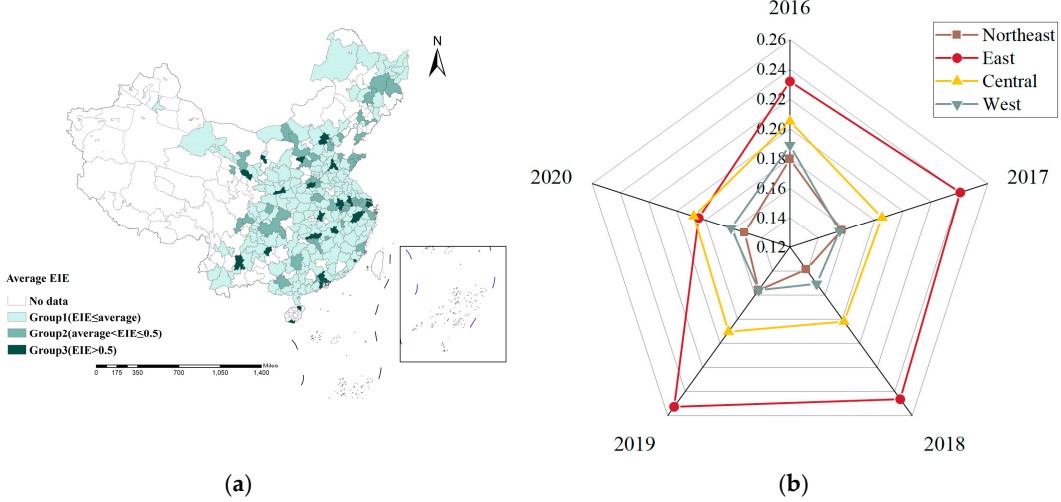

(**a**)        (**b**)

**Figure 8.** Spatial distribution characteristics of EIE: (**a**) Distribution of the EIE mean on the map of China; (**b**) Evolution of the EIE mean in the four main regions.

With developed economic infrastructure, high-level research institutions, and a large number of innovative enterprises, the eastern region has a relatively high level of innovation. In recent years, the central region has actively promoted innovation development, attracting the building of some high-tech industries and R&D centers, and its innovation level has gradually increased. The western regions are the relatively less economically developed regions of China, and these regions face a large development gap and the challenge of inadequate infrastructure. The northeastern region used to be China's heavy industrial base, but in recent years it has faced pressure from industrial restructuring and economic decline. Although the northeastern region still has some important innovative enterprises and R&D institutions, the overall innovation level is relatively low.

### 4.2.3. Stage 3: Innovative Wellbeing Efficiency

The third stage of the internal transformation of EWP is the transformation of innovation inputs into welfare outputs, i.e., innovation wellbeing efficiency (IWE). The IWE of 248 cities for 2016–2020 was plotted as a scatter plot, as shown in Figure 9. It can be seen that the mean value of IWE for 2016–2020 remained at a high level of about 0.6, and the cities that reached IWE effectiveness are 13, 6, 7, 9, and 2, respectively.

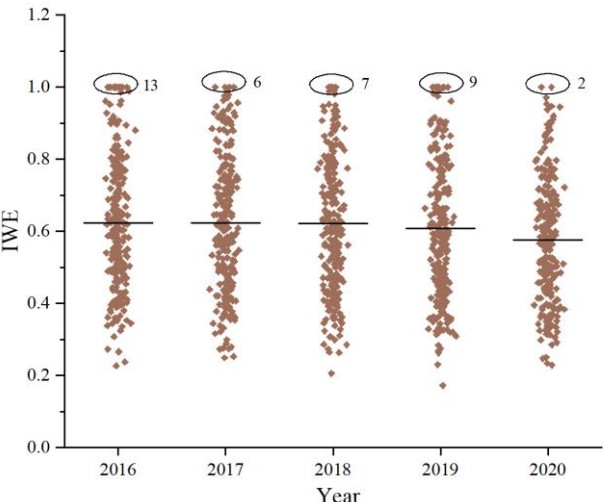

**Figure 9.** Scatter plot of IWE for 248 cities from 2016 to 2020.

This result is a profound indication that the Chinese government has recognized that the ultimate goal of development is comprehensive economic, social, and human development. Economic, social, scientific, and technological development is a means to achieve human development rather than a goal. China's all-around economic growth has contributed to a general improvement in people's living standards, improved nutrition, housing, and increased consumption and spending on education and health care.

The average values of IWE for 248 cities were plotted on a map of China, as shown in Figure 10a. All cities are classified into three groups, which are indicated by different colors: Group 1 (IWE ≤ average), Group 2 (average < IWE ≤ 0.8), and Group 3 (IWE > 0.8). In addition, the average values of IWE for 2016–2020 for the four major regions of China are plotted as radar plots (Figure 10b). As can be seen in Figure 10, the level of IWE in the east and central regions is much higher than that in the west and northeast areas, and there is a clear polarization trend.

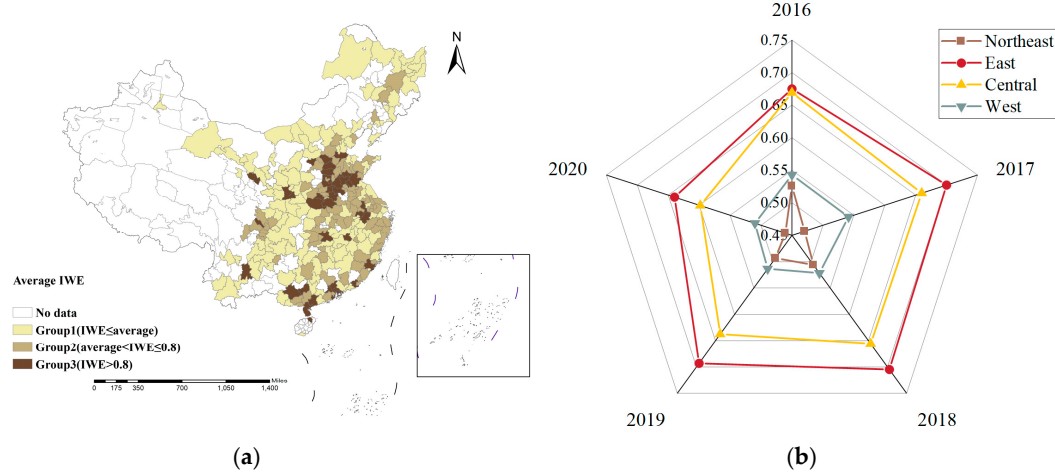

**Figure 10.** Spatial distribution characteristics of IWE: (**a**) Distribution of the IWE mean on the map of China; (**b**) Evolution of the IWE mean in the four main regions.

On the one hand, the eastern and central areas, as the pioneering regions of reform and opening up, have received more policy support and incentives, including tax concessions and industrial support. These policies help promote the economic development of the eastern and central regions. On the other hand, the eastern and central regions have invested more resources in infrastructure development. This includes the building of

transportation, energy, communication, and water conservancy. Good infrastructure helps to promote economic development, attract investment, and provide public services, and plays a key role in enhancing human wellbeing. The economic strength and well-developed infrastructure have made the eastern and central regions far superior to the western and northeastern regions in terms of education level and healthcare. In recent years, the Chinese government has taken a variety of measures to boost the development of the western and northeastern regions, including the Western Development Strategy and the Northeast Revitalization Plan. These policies aim to balance regional development, reduce regional disparities, and raise the level of human welfare in the western and northeastern regions.

## 5. Discussion

EWP represents the ultimate goal of sustainable development, and the enhancement of EWP is of great significance to human development. To comprehensively measure the current status and stage characteristics of sustainable development, this paper decomposes EWP internally into three sub-processes and empirically analyzes 248 prefecture-level cities in China using a network DDF model. The findings show consistency with previous research in the literature, as well as some new and interesting findings. A brief discussion of the findings is presented below.

The first finding is that EWP is at a low level in most Chinese cities. This result coincides with the studies of Yao et al. [16] and Zhou et al. [17]. Overall, the EWP situation in China is worrisome. The second finding is that, in terms of the performance of the three sub-stages of EWP, the best performance is in the first stage, followed by the third stage, and the worst is in the second stage. This suggests that the reason for China's overall poor EWP performance is the inefficient transformation of economic inputs into innovation outputs in the second stage. Studies have confirmed that China has long faced a lack of investment in STI talent and the STI environment [38], especially in resource-based cities in western and northeastern China [39]. The third is the discovery that, at the regional level, the higher-level EWPs are mostly clustered in the Yangtze River Delta, Pearl River Delta, Yellow River Delta, and Beijing-Tianjin-Hebei city clusters. This phenomenon can be corroborated by the separate studies of Li et al. [15], Hu et al. [58,74,75], Zhu et al. [59], and Xia and Li [60] for these four regions. Thus, EWPs form a gradually decreasing pattern in the eastern, central, western, and northeastern areas. The studies by Bian et al. [11] and Li [57] also showed that EWP performed better in eastern than in central and western China. But the stage performance varies among regions. The eastern region has lower EEE, but both EIE and IWE perform better than the national average; the central region is well-balanced between the three stages; the western region leads the country in EEE, but lags in both EIE and IWE; and the northeast region has inferior performance in all stages.

Given the results of this study, policy recommendations are provided to improve EWP in China's four major regions. In the eastern region, future development will prioritize the creation of a sustainable, greener, and harmonious ecological environment. Such efforts include intensifying pollution prevention and control measures, vigorously developing clean energy sources, and improving ecological and environmental protection awareness. In the Northeast, the objective of future development is to attain high-quality, sustainable, and dynamic economic development. Given the Northeast's economy's persistent downward pressure, caution should be exercised against resuming an outdated "growth-first" philosophy that may allocate more resources to archaic industries instead of more competitive and service-oriented industries that align more favorably with China's robust economic development. In the central region, the focus of future development should be to provide better balanced, adequate, and high-quality public services. Examples include setting up a diverse health and elderly service system that caters to the various needs of the aging population. Establishing a full life-cycle public service system ranging from preschool to higher education can further improve residents' scientific literacy, cultural quality, and ability to lead successful lives. In the western region, development efforts should focus on two fronts. Firstly, there is a need to increase the momentum of innovative

development to improve the level of industrial structure given the region's small economic scale and low level of industrial structure. Secondly, efforts are needed to narrow the gap between rural and urban areas within the region. Such modernization typically results in a substantial divergence between urban and rural areas, posing an overall challenge to the western region.

The theoretical and practical contributions of the study are as follows. (1) It opens up the transformation process within EWP, innovatively decomposes EWP into three stages: ecological economic efficiency, economic innovation efficiency, and innovation wellbeing efficiency, and incorporates natural resource consumption, economic growth, science and technology innovation and people's welfare, which have been widely studied, into the same urban system, providing a new research paradigm. (2) A three-stage network evaluation model of EWP and a set of applicable evaluation index systems are constructed, expanding the previous studies. (3) Based on the research results, targeted policy recommendations are proposed for the four major regions of China to enhance EWP. At the same time, since the resource endowments and economic situations of the four major regions in China are different and very representative, the policy recommendations in this paper have important reference values for other countries and regions in the world to enhance EWP.

However, there are some areas for improvement in this paper. On the one hand, China is a large country, and although dividing China into four regions can better take into account regional differences, it is somewhat biased for some cities. On the other hand, when new indicator data appear, the indicator system has to be updated accordingly, and the results obtained may also deviate from this paper.

## 6. Conclusions

EWP enhancement is critical for China to achieve more sustainable economic development and human development. This paper proposes a three-stage network efficiency evaluation model to decompose EWP into three stages, namely ecological economic efficiency (EEE), economic innovation efficiency (EIE), and innovation wellbeing efficiency (IWE). The directional distance function (DDF) model was utilized to assess the overall EWP efficiency and sub-stage efficiency of 248 cities between 2016 and 2020. The study results indicate that EWP in China is generally low. In terms of the three sub-stages of EWP, the first stage performs the best, the third stage performs the second best, and the second stage is the worst. The research suggests that the principal reason for the generally low EWP may be linked to the conversion efficiency of economic inputs into innovative outputs in the second stage. From a regional viewpoint, EWP generally shows a gradual decreasing trend from the east, central, and west to the northeast, but the stage performance varies among regions. The eastern region has lower EEE, but both EIE and IWE perform better than the national average; the central region is well-balanced between the three stages; the western region leads the country in EEE, but lags in both EIE and IWE; and the northeast region has inferior performance in all stages. This study's findings provide vital reference values for policymakers to determine key points for enhancing EWP in different regions of China.

**Author Contributions:** The authors confirm their contribution to the paper: Y.Z. writing—original draft preparation, visualization, and formal analysis. X.C. writing—original draft preparation, conceptualization, data curation, visualization, supervision. Y.M. methodology, formal analysis, software, visualization. L.J. conceptualization, supervision, writing—reviewing and editing. L.W. conceptualization, supervision. All authors have read and agreed to the published version of the manuscript.

**Funding:** This work is supported by the National Natural Science Foundation of China (Grant No. 72204033), the Humanities and Social Science project of Ministry of Education of China (Grant No. 21YJC630169), the China Postdoctoral Science Foundation (Grant No. 2022M711457), the Natural Science Foundation of Chongqing (Grant No. cstc2021jcyj-msxmX1010 and CSTB2022NSCQ-MSX0390), the Social Science Planning Project of Chongqing (Grant No. 2020QNGL25), and the Humanities and Social Science Research Project of Chongqing Education Commission (Grant No. 21SKJD072).

**Institutional Review Board Statement:** Not applicable.

**Informed Consent Statement:** Not applicable.

**Data Availability Statement:** All data included in this study are available upon request by contacting the corresponding author.

**Conflicts of Interest:** The authors declare no competing interest.

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
