# Peer review of "What Is the State of Development of Eco-Wellbeing Performance in China? An Analysis from a Three-Stage Network Perspective"

_land, doi:10.3390/land12081512_

Round 1

Reviewer 1 Report

I am honored to review this paper, i have following suggestions:

1. The introduction section can be shortened to cut into the research topic more quickly.

2. The literature review should focus on the relationship between the three stages, explain the logic of their evolution, how each stage is inherited, and how to feedback.

3. In terms of evaluation index system, there is a mature evaluation system for the evaluation of innovation efficiency, such as R&D personnel input, R&D funds input and other indicators. How do you use other indicators here.

4. The conclusion can be written in separate paragraphs.

Author Response

Response to Reviewer 1 Comments

Point 1: The introduction section can be shortened to cut into the research topic more quickly.

Response 1: Thanks for your constructive suggestions.

The background of the study in the introduction section has been improved and shortened in the revised manuscript to move more quickly to the topic of the study.

Point 2: The literature review should focus on the relationship between the three stages, explain the logic of their evolution, how each stage is inherited, and how to feedback.

Response 2: Thanks a lot for your valuable comment.

A new section 2.4 has been added in the revised manuscript to provide a more detailed description of the linkage and evolutionary logic of the three stages.

“All of the above studies take ecological inputs to cities as a starting point and measure their efficiency in translating into economic output, innovation output, and people's well-being, respectively. However, the correlation between the economy, innovation, and people's well-being has been ignored. What is certain is that innovation efficiency is an upgrade of eco-efficiency and eco-wellbeing performance is an upgrade of innovation efficiency. The emergence of innovation efficiency has led to a shift away from a focus on crude economic growth, and the introduction of eco-wellbeing has anchored the purpose of development in human beings themselves. Today, there is a consensus that cities are complex mega systems [62]. Therefore, the depletion of natural resources, economic development, technological innovation, and the well-being of people in cities can never exist separately from this system. It wasn't until Kiani Mavi et al. [63] used the two-stage network DEA to measure eco-efficiency and eco-innovation that a new way of thinking was opened to clarify this connection. Since then, many scholars have started to conduct studies using this approach, for example, Hou et al. [64] and Xia & Li [60] decomposed EWP into ecological economic efficiency and economic welfare efficiency. However, they usually only remain in a two-stage research paradigm [65-68]. The closest to the idea of this study is that of Zhang et al. [69] who measured eco-efficiency, eco-technological innovation, and eco-welfare performance in 102 countries, however, they still failed to link all three into the same framework. The final objective of urban development is the welfare of the people, while technological progress and innovation are only an intermediate process. Therefore, this paper innovatively decomposes EWP into three interrelated stages, namely, the conversion of ecological inputs into economic outputs, economic inputs into innovation outputs, and innovation inputs into welfare outputs. For ease of presentation, they are denoted by ecological economic efficiency (EEE), economic innovation efficiency (EIE), and innovation well-being efficiency (IWE), respectively.”

Point 3: In terms of evaluation index system, there is a mature evaluation system for the evaluation of innovation efficiency, such as R&D personnel input, R&D funds input and other indicators. How do you use other indicators here.

Response 3: Thanks a lot for your insightful comments.

We strongly agree that R&D personnel and R&D expenditure, etc. are valid indicators to measure innovation efficiency. In this study, we also retain this idea to construct the evaluation index system of innovation efficiency. Unfortunately, the data for R&D personnel and R&D expenditure are also unavailable at city level in China. Therefore, referring to Wu and Liu [73], we choose the number of higher education teachers as a proxy indicator for R&D personnel, and referring to Wang et al. [68], we choose the expenditure of science and technology as a proxy indicator for R&D expenditure. The number of patents is an indicator that has been widely used to measure the efficiency of innovation in a region [44].

Point 4: The conclusion can be written in separate paragraphs.

Response 4: The authors appreciated for your valuable suggestion.

The conclusions in Section 6 have been singled out in the revised manuscript. And, Section has been added to discuss the findings, policy implications, contributions, and shortcomings of this study.

Reviewer 2 Report

 1.       The authors started the introduction from the background of China society, as an international academic paper, it will be better to start from the global context and international policy or academic background.

2.       The authors should provide a clear research purpose or hypothesis between the 2. Literature review and 3. Methodology and data sources sections. Lacking of a clear research purpose and hypothesis, it is difficult to realize whether your data collection and research method are feasible while the results are reasonable or not.

3.       A rational explanation about why choosing Network-DDF and why directional distance function (DDF) model is suitable for your research should be clearly stated.

4.       A brief definition Northeast, Eastern, Central and Western regions is needed for the explanation of Figures 3-9.

5.       Totally, the image quality of all figures seems low, please replace them with higher quality version which are more readable.

6.       A discussion section is suggested to be added before the conclusion section. Concretely, a full-scale discussion section is considered essential to discuss and explore your finding based on the existing finding of previous studies to clarify your contribution for the international context.

The paper must be professionally edited for expression especially after the major revision of the paper.

Author Response

Response to Reviewer 2 Comments

Point 1: The authors started the introduction from the background of China society, as an international academic paper, it will be better to start from the global context and international policy or academic background.

Response 1: Thanks to the reviewer for the suggestion. In the revised manuscript, the introduction has been improved to introduce the theme of this study in a global context.

“Today's world is undergoing significant and unprecedented changes. Global climate change, social unrest due to the growing inequality between certain countries, and the impact of the fourth technological and industrial revolution on the labor market are all severe obstacles to humanity's goal of a better life [1]. At the same time, China, as the world's most populous country, has quietly changed its main social contradictions, and its unbalanced and inadequate development can no longer meet the people's growing demand for a high quality of life [2].”

Point 2: The authors should provide a clear research purpose or hypothesis between the 2. Literature review and 3. Methodology and data sources sections. Lacking of a clear research purpose and hypothesis, it is difficult to realize whether your data collection and research method are feasible while the results are reasonable or not.

Response 2: The authors thanks for your kind suggestion. The purpose of this research has been added at the end of section 2.3 of the revised manuscript.

“The purpose of this paper is to construct an appropriate set of models and indicators to measure the overall and stage efficiency of EWP in 248 Chinese cities for the period 2016-2020. By analyzing the performance of each sub-stage, the key aspects affecting the EWP are identified, and thus specific areas for policy improvement are suggested.”

Point 3: A rational explanation about why choosing Network-DDF and why directional distance function (DDF) model is suitable for your research should be clearly stated.

Response 3: Thank you for the kind advice. The detailed explanation about why choosing Network-DDF and why DDF model is suitable for the research have been added in Section 3.1 of the revised manuscript.

“Although there are many methods to measure efficiencies, such as the ratio method and stochastic frontier analysis, DEA has become the mainstream method nowadays due to its advantages of handling multiple inputs and outputs and not requiring pre-set parameters [54,60]. Meanwhile, among various DEA extension models, radial models, slack-based measure (SBM), and directional distance function (DDF) are the three most widely used ones. However, the radial DEA model cannot directly deal with undesired outputs and therefore cannot effectively balance the two objectives of minimizing undesired outputs and maximizing desired outputs [11]. Although the SBM model can deal with undesired outputs well, has the characteristic that more input and output indicators are better [71], which tends to underestimate the efficiency in the actual situation of this study. Chung et al. [72] proposed the DDF model which allows the simultaneous reduction of inputs and non-desired outputs as well as increasing desired outputs without underestimating efficiency in the case of less desired outputs. Therefore, in this paper, the DDF is used to measure the overall and phased efficiency of the three-stage network.”

Point 4: A brief definition Northeast, Eastern, Central and Western regions is needed for the explanation of Figures 3-9.

Response 4: The authors thanks for your kind suggestion. The detailed definition of Northeast, Eastern, Central and Western regions has been added in Section 3.3. And, Figure 2 has been added to explain the definition and division of those four major regions of China.

“China is a large country with a massive population and a vast land area, and different regions have large differences in various aspects such as natural ecology and social environment. For subsequent analysis, this paper adopts a spatial perspective and divides mainland China into four major zones according to the distribution of natural resources and economic and social development: the eastern, central, western, and northeastern areas [74]. As shown in Figure 2, 10 regions such as Beijing and Tianjin belong to the east, 6 regions such as Hubei and Hunan belong to the center, 12 regions such as Sichuan and Chongqing belong to the west, and the northeast contains 3 provinces of Liaoning, Jilin, and Heilongjiang.”

Point 5: Totally, the image quality of all figures seems low, please replace them with higher quality version which are more readable.

Response 5: We are extremely grateful to the reviewer for pointing out this issue.

The resolution of all images in the revised manuscript has been significantly improved.

Point 6: A discussion section is suggested to be added before the conclusion section. Concretely, a full-scale discussion section is considered essential to discuss and explore your finding based on the existing finding of previous studies to clarify your contribution for the international context.

Response 6: The authors appreciated for your valuable suggestion. A new section 5 has been added to the revised manuscript, which specifically includes: (1) A discussion of the findings of this study based on existing research. (2) Policy implications. (3) Discussion of the contributions and limitations of this study in an international context.

“EWP represents the ultimate goal of sustainable development, and the enhancement of EWP is of great significance to human development. To comprehensively measure the current status and stage characteristics of sustainable development, this paper decomposes EWP internally into three sub-processes and empirically analyzes 248 prefecture-level cities in China using a network DDF model. The findings show consistency with previous literature, as well as some new and interesting findings. A brief discussion of the findings is presented below.

The first finding is that EWP is at a low level in most Chinese cities. This result coincides with the study of Yao et al. [16] and Zhou et al. [17]. Overall, the EWP situation in China is worrisome. The second finding is that in terms of the performance of the three sub-stages of EWP, the best performance is in the first stage, followed by the third stage, and the worst is in the second stage. This suggests that the reason for China's overall poor EWP performance is the inefficient transformation of economic inputs into innovation outputs in the second stage. Studies have confirmed that China has long faced a lack of investment in STI talent and the STI environment [38], especially in resource-based cities in western and northeastern China [39]. The third is the discovery at the regional level, the higher-level EWPs are mostly clustered in the Yangtze River Delta, Pearl River Delta, Yellow River Delta, and Beijing-Tianjin-Hebei city clusters. This phenomenon can be corroborated by the separate studies of Li et al. [15], Hu et al. [58,75,76], Zhu et al. [59], and Xia & Li [60] for these four regions. Thus, EWPs form a gradually decreasing pattern in the eastern, central, western, and northeastern areas. The studies by Bian et al. [11] and Li [57] also showed that EWP performed better in eastern than in central and western China. But the stage performance varies among regions. The eastern region has lower EEE, but both EIE and IWE perform better than the national average; the central region is well-balanced between the three stages; the western region leads the country in EEE, but lags in both EIE and IWE; and the northeast region has inferior performance in all stages.

Given the results of this study, policy recommendations are provided to improve EWP in China's four major regions. In the eastern region, future development will prioritize the creation of a sustainable, greener, and harmonious ecological environment. Such efforts include intensifying pollution prevention and control measures, vigorously developing clean energy sources, and improving ecological and environmental protection awareness. In the Northeast, the objective of future development is to attain high-quality, sustainable, and dynamic economic development. Given the Northeast's economy's persistent downward pressure, caution should be exercised against resuming an outdated "growth-first" philosophy that may allocate more resources to archaic industries instead of more competitive and service-oriented industries that align more favorably with China's robust economic development. In the central region, the focus of future development should be to provide better balanced, adequate, and high-quality public services. Examples include setting up a diverse health and elderly service system that caters to the various needs of the aging population. Establishing a full life-cycle public service system ranging from preschool to higher education can further improve residents' scientific literacy, cultural quality, and ability to lead successful lives. In the western region, development efforts should focus on two fronts. Firstly, there's a need to increase the momentum of innovative development to improve the level of industrial structure given the region's small economic scale and low level of industrial structure. Secondly, efforts are needed to narrow the gap between rural and urban areas within the region. Such modernization typically results in a substantial divergence between urban and rural areas, posing an overall challenge to the Western region.

The theoretical and practical contributions of the study are as follows: (1) It opens up the transformation process within EWP, innovatively decomposes EWP into three stages: ecological economic efficiency, economic innovation efficiency, and innovation wellbeing efficiency, and incorporates natural resource consumption, economic growth, science and technology innovation and people's welfare, which have been widely studied, into the same urban system, providing a new research paradigm. (2) A three-stage network evaluation model of EWP and a set of applicable evaluation index systems are constructed, expanding the previous studies. (3) Based on the research results, targeted policy recommendations are proposed for the four major regions of China to enhance EWP. At the same time, since the resource endowments and economic situations of the four major regions in China are different and very representative, the policy recommendations in this paper have important reference values for other countries and regions in the world to enhance EWP.

Round 2

Reviewer 2 Report

I would like to thank the authors for their great effort. Basically, the revisions conducted by the authors are acceptable.

Minor editing of English language required.